# Ambient pollution at hip fracture units and impact on mortality and post-operative delirium: A hormetic effect?

**Chika Edward Uzoigwe**[1]*, **Rana Muhammad Anss Bin Qadir**[2], **Ahmed Daoub**[3]

1 Harcourt House, Sheffield, United Kingdom, 2 Cardiff and Vale University Health Board, Cardiff, United Kingdom, 3 Robert Jones & Agnes Hunt Orthopaedic Hospital, Gobowen, United Kingdom

* chika@doctors.org.uk

**Data Availability Statement:** All relevant data are within the manuscript and its Supporting Information files.

## Abstract

There is increasing awareness of the deleterious effects of ambient pollution. The World Health Organisation (WHO) has recently advocated new safe limits of annual exposure for the three pre-dominant pollutants: fine particulate matter ($PM_{2.5}$), coarse particulate matter ($PM_{10}$) and nitrogen dioxide; namely $5\mu g/m^3$, $15\mu g/m^3$ and $10g\mu/m^3$ respectively. Both the USA and UK have recently implemented news standards which are lower than their current values, but still exceed those espoused by WHO. The WHO thresholds are challenging targets. It remains to be determined the proportion of secondary healthcare institutions located in zones with mean ambient pollutant levels in excess of the WHO limits and the impact this has on patients treated at these centres. This is particularly so for elderly patients who are theoretically most vulnerable to the adverse sequel of pollutant exposure. Using the UK National Hip Fracture Database and Defra Data (Department of Environmental, Food & Rural Affairs) we determined the annual mean $PM_{2.5}$ $PM_{10}$ and nitrogen dioxide exposure for all the units treating senescent hip fracture patients. We correlated these ambient pollutant levels with all-cause 30-day mortality and incidence of post-operative delirium for hip fracture patients. The vast majority (96%) of hip fracture units were located in zones where mean $PM_{2.5}$ levels exceeded that required by the WHO guidance. A sizeable proportion also had annual mean exposures that surpassed the WHO $PM_{10}$ (14.8%) and nitrogen dioxide (63%) recommended thresholds. There was no difference in 30-day mortality between hip fracture patients treated at units located in areas where pollutant titres were subliminal to the WHO guidance levels and those treated at centres where WHO thresholds were exceeded. By way of contrast patients admitted to institutions with mean ambient $PM_{10}$ and nitrogen dioxide concentrations that surpassed the WHO limits had a lower risk of post-operative delirium compared to those at centres where the mean levels did not breach the WHO limit. For $PM_{10}$ the relative risk was 0.89 CI:0.82–0.92 (p<0.0001) and that for nitrogen dioxide 0.92 CI: 0.89–0.94 (p<0.0001). The WHO target is ambitious as it relates to healthcare institutions. The majority are in areas that exceed WHO recommended limits. This does not appear to impact upon mortality for hip fracture patients. The decrease in incidence in post-operative delirium in areas of higher exposure raises, again in an epidemiological study, the possibility of the enigmatic phenomenon of hormesis, an adaptive response whereby low-

**Funding:** The author(s) received no specific funding for this work.

**Competing interests:** The authors have declared that no competing interests exist.

dose exposure to a noxious agent or physiological stress enhances future physiological function.

## Introduction

Climate and environmental pollution represent two new challenges to healthcare almost peculiar to the 21[st] century. It is well established that atmospheric pollutants have a direct impact on health both acutely and in the long-term [1–3]. Effects include increased risk of cardiovascular, respiratory and adverse neurocognitive effects. Indeed this is so much so that it is thought the dramatic decline in cardiovascular events during COVID lockdowns may be attributable to a reduction in traffic and attendant decline in atmospheric pollutants [4]. The pollutants are divided into particulate matter (PM) and noxious gases. Particulate matter is further divided into subgroups contingent upon the maximum size of the particles. Fine or $PM_{2.5}$ pollutants comprise particles less than 2.5μm and coarse or $PM_{10}$ refers to particulate matter of maximum size 10μm. The most significant pollutant gas is nitrogen dioxide ($NO_2$).

According to UK legislation, The Air Quality Standards Regulations 2010, annual mean concentration of $PM_{2.5}$ should not exceed 20μg/m$^3$ [5]. That of $PM_{10}$ should not surpass 40μg/m$^3$. The legislative limit for nitrogen dioxide is 40μg/m$^3$. The most recent limits recommended by the World Health Organisation in their 2021 guidance for $PM_{2.5}$ and $PM_{10}$ are considerably lower at 5μg/m$^3$ and 15μg/m$^3$ respectively [6]. WHO ideal upper limit for mean annual exposure to $NO_2$ is 10μg/m$^3$. In response to the new WHO guidance both the UK and the USA have sought to reduce their statutory and policy thresholds [7].

WHO in their target level accompanying publication perform a review of the literature. They show robust evidence linking pollutants to mortality and mortality. This expressly included neurocognitive endpoints [6]. The main point of contention is at what thresholds exposures are injurious and/or if there is any level below which exposure engenders no harm.

There is a need for hospitals to be accessible. For this reason they tend to be situated at the confluence of transport networks and at a focal point of residential nexus. A study by British Lung Foundation reported that third of GP practices and a quarter of hospitals were in areas exceeding WHO 2018 pollution limits [8]. The study looked only at $PM_{2.5}$ using the historical 2018 WHO standard for $PM_{2.5}$ of 10μg/m$^3$. No relation was made to the UK legal limit. No specific outcomes were explored. The question of the impact of ambient pollutions on acutely hospitalised cohort remains unanswered. This is particularly so in the case of vulnerable and elderly patients who fall within the demographic of those at risk from the very health conditions whose risk is amplified by pollution. Interventions to reduce ambient pollutions can dramatically effective in not expensive [9]. However one must attend to level of reduction necessary if any. We sought to determine the ambient pollutions around hip fracture units and its relation to current UK and WHO standards and to outcomes with regard to 30-day mortality and post-operative delirium. This patient cohort was specifically selected due their senescence, reduced physiological reserve and thereby a group likely to most acutely vulnerable to ambient levels of pollutions and whom the implemented regulations should protect.

## Methods

We performed a cross-sectional observational study. The UK National Hip Fracture Database is one of the largest audits of its kind (https://www.nhfd.co.uk/). It records the clinical outcomes of all hip fracture patients, over 60 years of age, presenting to institutions in England

and Wales. It thus reports institutional level outcome figures including mortality and incidence of delerium We identified all units treating hip fracture patients from the most recent complete National Hip Fracture Database year 2022. Using the address of the hospitals we calculated the mean annual values for $PM_{2.5}$, $PM_{10}$ and nitrogen dioxide using meteorological data from the UK Government Department for Environment Food and Rural Affairs (Defra) [10]. We sought to determine if there was any relation with mortality and the prevalence of post-operative delirium. Post-operative delirium was recorded in the National Hip Fracture Database for the year 2022. For each institution we used the latest mortality recorded for that institution in the National Hip Fracture Database Mortality figures which were from 2019. We compared unit exposures with WHO limits. We compared the outcomes hip fracture centres located with mean ambient $PM_{2.5}$, $PM_{10}$ and nitrogen dioxide levels above WHO standards with those in zones beneath this threshold.

The Strengthening the Reporting of Observational Studies in Epidemiology were followed.

## Statistical methods

Proportions were compared with Chi-squared Test.

## Results

In the year 2022 there were 168 units treating hip fracture. There was no pollution data recoverable for those on mainland Northern Ireland, Isle of Man, or Guernsey. These were thus excluded. The remaining units numbered 162. There were a total of 69,461 patients treating in these units. The numbers of patients treated in centres ranged from 17 to 1058 with an average of 429.

### $PM_{2.5}$

Range of $PM_{2.5}$ concentrations was 4.4μg/m$^3$ to 11.2μg/m$^3$ (Table 1). The mean annual ambient concentration for $PM_{2.5}$ was below the WHO limit for only 5 (3.1%) of the 162 institutions. Two of these were in Wales and the remaining the North East or North West. For the remaining 157 (96.9%) institutions, mean levels exceeded WHO's healthy limit. Indeed for 11 (6.8%) double the limit was exceeded. As corollary to this 97.7% of patients were treated in units exposed to environmental pollution that exceeded WHO safe-limits and 4.2% at units that exceeded double the limit. None exceeded the UK legislative limit of 20μg/m$^3$.

### $PM_{10}$

Mean annual $PM_{10}$ concentrations at hip fracture units ranged from 7.7μg/m$^3$ to 18.7μg/m$^3$ (Table 1). 138 (85.2%) of institutions were situated in areas with levels subliminal to the WHO

**Table 1. Exposures to $PM_{2.5}$, $PM_{10}$ and Nitrogen dioxide.**

|  | $PM_{2.5}$ | $PM_{10}$ | Nitrogen Dioxide |
|---|---|---|---|
| **WHO Limit μg/m$^3$** | 5 | 15 | 10 |
| **Mean μg/m3** | 7.5 | 12.8 | 12.4 |
| **Range μg/m3** | 4.4–11.2 | 7.7–18.7 | 3.6–32.4 |
| **Institutions above threshold** | 96.9% | 14.8% | 63.0% |
| **Patients treated at units above threshold** | 97.7% | 10.1% | 62.9% |
| **Institutions above double the limit** | 6.8% | 0% | 8.6% |
| **Patients treated at units above double the limit** | 4.2% | 0% | 5.4% |

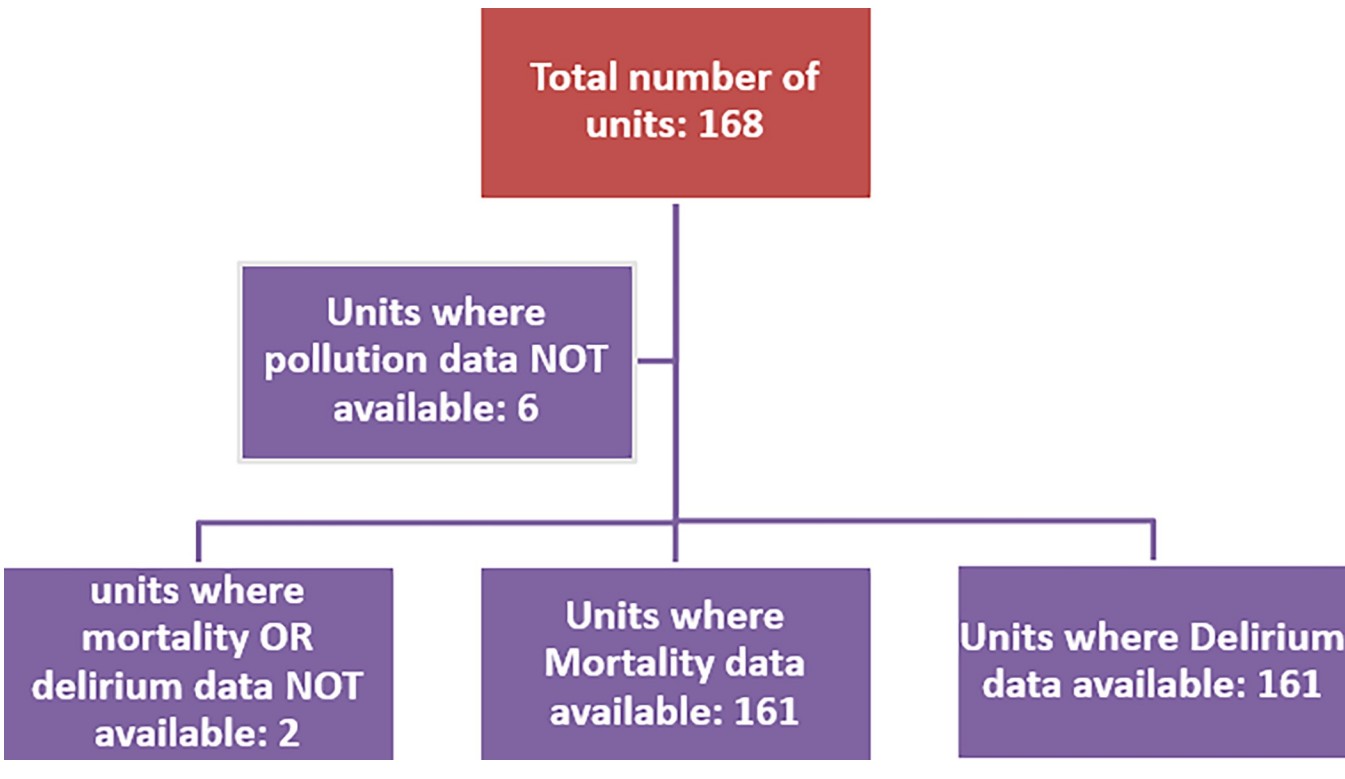

**Fig 1. Numerical distribution of units.**

threshold of health. The remainder 24 hip fracture units (14.8%) suffered mean exposure in excess of the recommended limit. This involved 10.1% of the 69,461 hip fracture patients included in this works. Ambient pollution did not surpass the UK legal mean annual limit for any unit.

## Nitrogen dioxide

$NO_2$ concentrations varied from 3.6μg/m³ to 32.4μg/m³. 60 (37.0%) units were had annual exposure beneath the limit for WHO (Table 1). Hence 102 (63.0%) the mean ambient concentration surpassed the limit. 62.9% of hip fracture patients were treated at these units. 14 (8.6%) units had mean annual atmospheric titres that exceed double the safe limit. These centres were responsible for 5.4% of hip fracture patients. At no unit was the statutory limit exceeded.

## Mortality

We used the latest mortality figures for each institution was derived from 2019 (Fig 1). Data was available for 160 units of the 162 included. Overall mortality was 6.5% with an institutional range of 2% to 11%. Mean annual ambient pollution concentration at hip fracture units whether above or the WHO standard did not impact upon mortality (Table 2).

## Delirium

Data on post-operative delirium was reported in 161 of the 162 included hip fracture units. The group mean was 29% with an institutional range of 1.5% to 99.3% (Fig 1) (Table 3). For $PM_{2.5}$, the rate of post-operative delirium was unrelated to mean ambient hospital levels above

**Table 2. Mortality of patients treated at units above and below the WHO standard.**

| | $PM_{2.5}$ | $PM_{10}$ | Nitrogen Dioxide |
|---|---|---|---|
| Mortality sub-WHO threshold (%) | 5.6 | 6.6 | 6.7 |
| Mortality supra-WHO threshold (%) | 6.6 | 6.6 | 6.5 |
| Mortality above x2 WHO threshold (%) | 6.0 | N/A | 6.0 |
| Mortality Sub vs supra WHO threshold Relative Risk (CI) (p-value) | 1.2(0.94–1.4) (0.16) | 0.98 (0.93–1.0) (0.9) | 0.98(0.92–1.0) (0.43) |
| Mortality Sub vs x2 supra WHO threshold Relative Risk (CI)(p-value) | 1.1 (0.83–1.4) (0.65) | N/A | 0.91(0.79–1.0) (0.18) |

or below the WHO recommended ceiling level. This was not so for $PM_{10}$ and nitrogen dioxide. For these pollutants the incidence of post-operative delirium was actually lower for those receiving treatment for hip fracture at units with higher mean annual titres of pollutant.

## Discussion

We looked at the annual pollutant exposure at hip fracture units in England and Wales. Levels of the $PM_{2.5}$ exceeded the WHO limit at the vast of centres (96.9%) and exceeded double the limit a non-negligible proportion (6.8%). A notable proportion of centres were also located in areas where mean titres of $PM_{10}$ and nitrogen oxide surpassed the WHO threshold (14.8% and 63%) respectively.

We used the latest available figures from the NHFD as estimates for mortality from 2019. We did not observe any difference in mortality between hip fracture institutions located in areas with mean annual ambient pollution levels below the WHO limits and those in locations above such thresholds. This held for all pollutants, $PM_{2.5}$, $PM_{10}$ and nitrogen dioxide.

Unexpectedly we did however observe that the incidence of post-operative delirium was lower at units where environmental air pollution exceeded WHO recommended threshold for air safety for $PM_{10}$ (RR: 089 CI: 0.85–92 p<0.0001) and Nitrogen Dioxide (RR 0.92: CI 0.89-.094 p<0.0001). This was not the case for $PM_{2.5}$. The finding raises the possibility of the enigmatic but well-characterised phenomenon of hormesis [11,12]. Hormesis is an adaptive response whereby exposure to low, non-injurious doses, of an otherwise noxious agent, or to other biological challenges actually results in improved physiological function especially to the originator provoking agent or challenge. The classical example is ischaemic pre-conditioning of the heart. Here transient episodes of ischaemia render the heart more resistant to future more protracted or extreme episodes of ischaemia [13]. This effect can be exploited. However exercise and intermittent fasting and caloric restriction are also forms "physiological hormesis" with a host of health benefits.

Hormesis is highly evolutionary preserved phenomenon found through the whole gamut of taxa from antiquity to today. It is not phenomenon new to the 21$^{st}$ century nor exclusive to

**Table 3. Delirium of patients treated at units above below WHO standard.**

| | $PM_{2.5}$ | $PM_{10}$ | Nitrogen Dioxide |
|---|---|---|---|
| Delirium sub-WHO threshold (%) | 29.7 | 29.5 | 31.2 |
| Delirium above-WHO threshold (%) | 29.0 | 25.3 | 27.8 |
| Delirium above x2 WHO threshold (%) | 29.3 | N/A | 29.7 |
| Delirium Sub vs supra-WHO threshold Relative Risk (CI)(p-value) | 0.98(0.91–1.1) (0.67) | **0.89(0.85–0.92) (<0.0001)** | **0.92(0.89–0.94) (<0.0001)** |
| Delirium Sub vs x2 supra-WHO threshold Relative Risk (CI)(p-value) | 0.99(0.90–1.1) (0.86) | N/A | 0.96 (0.91–1.0) (0.19) |

human or sophisticated animal physiology. Given that is so pervasive it is bound to have ramifications and manifestations in the healthcare setting. Indeed hormesis was first observed in plants [14]. Here plants cultivated in soils with low concentrations of pollutant metals ions, which were injurious at higher doses, actually grow faster than those germinated in soils without these chemicals. Hormesis was then observed in human epidemiology specifically in the area of air pollution and radiation in Phalen's classic paper 20 years ago [15]. This is a finding consistently reproduced [16]. Villeneuve in his large study, of comparative size to ours, including almost 90,000 women, found that those with the lowest non-accidental mortality were those who with mean $PM_{2.5}$ exposures of around $7\mu g/m^3$ [17]. As exposure to this pollutant fell below this level mortality began to rise. As exposure exceeded this level mortality also rose but more steeply. Hormesis thus produces the classical "J" or "U" shaped curve. The phenomenon was then observed again in medical practice with regard to radiation exposure [18,19]. Radiation exposures that are greater than background, but below critical thresholds, do not harm; but rather may be salubrious. This remains controversial as some bodies drive single-mindedly to eliminate all exposures to radiation, as high doses of are incontrovertibly deleterious.

Hormesis has also been observed to be operative in the risk of neurocognitive conditions. Tellingly in a large meta-analysis Han et al showed remote ischaemic preconditioning to reduce the risk of post-operative cognitive dysfunction (POCD) by up to 50% [20]. This is congruent with the hypothesis that transient perioperative hypo-perfusion and/or inflammation may be contributory to post-operative cognitive decline.

The established research corpus on hormesis is transferrable to our findings. The WHO $PM_{2.5}$ $PM_{10}$ and nitrogen dioxide thresholds are very low and indeed were derived from the lowest limits of exposure used in epidemiological studies [6]. Hence those with exposures sub-liminal to this may fall within the sub-hormetic range. Further in no institution did the mean annual ambient pollutant level exceed the UK legislative limit. A notable proportion of those above the WHO limit may putatively fall within the hermetic range. The UK National Hip Fracture Database includes only patients 60 years and greater in age. Hip fracture is generally an injury of senescence. Hence the fact that those included in the study have attained such an age suggests that they represent a cohort that is not acutely sensitive to pollutants so as suffer premature mortality as sequel to exposure to these pollutants. Hospitals in the UK only treat patients within their catchment area. Hence it is likely that the mean annual hospital pollutant exposure reflects that of the patients' residential milieu and indeed where this senescent cohort of spends much of their time. Perioperative hypoxia, physiological stress and possibly even anaesthetic agents eliciting "neuro-inflammation" are thought to be potentially contributory causes of postoperative delirium [21]. It is conceivable that previous low-dose exposure to pollutants, mimicking these effects to some degree, may render patients more resistant to subsequent episodes of hypo-perfusion, hypoxia and/or inflammation in the perioperative window. This is protective against early neurocognitive adverse effects of surgery. It is well-established that pollutants at high levels produce inflammatory type physiological challenges.

We noted that for $PM_{2.5}$ Villeneuve observed hermetic threshold of $7\mu g/m^3$. We therefore considered that our level of 5μg/m3 maybe too low; not only physiologically but equally statistically due to the sample size, as it included only 5 of the total institutions included in the study. We performed a retrospective analysis of the data using the limit of greater of $7\mu g/m^3$. This revealed a that a hermetic effect for $PM_{2.5}$ with regard to post-operative delirium. Hence those treated at institutions with $PM_{2.5}$ values greater than $7 \mu g/m^3$ had a lower risk of post-operative delirium (relative risk: 0.95 CI:0.93–0.97 p = 0.0001).

It is noteworthy however there was no differences in the incidence of post-operative delirium, between those treated at units with mean annual pollutant titres below WHO threshold

and where exposures were the highest, at double the WHO limit. Units are the highest exposure where thus is the putative supra-hormetic range.

Consistent with our work and the possibility of hermetic levels, Che et al conducted a study in China looking at the impact of mean daily nosocomial pollutant exposures and postoperative delirium [22]. In their study the mean ambient levels were astronomically high by UK standards. The mean exposure for $PM_{2.5}$, $PM_{10}$ and nitrogen dioxide were 64μg/m$^3$, 107μg/m$^3$ and 44μg/m$^3$ respectively. Essentially 3.5 to 9 fold higher than UK values and considerably surpassing WHO and the UK statutory limits. The authors found daily increases in $PM_{2.5}$ of 48μg/m$^3$; from the previous day, was associated with an increase in post-operative delirium. These authors are operating in pollutant levels an order of magnitude higher than the present study and clearly in the supra-hormetic zone. However the disparate findings support the hermetic possibility.

Shi et al also used the UK National Hip Fracture Database to correlate pollution with fracture incidence and mortality [23]. They found increasing $PM_{2.5}$ $PM_{10}$ and nitrogen dioxide levels were associated increased 30-day all-cause mortality of hip fracture patients. Each 5μg/m$^3$ increase in each agent was associated with a 9.3%, 8.3%, and 2.9% increase in 30-day mortality respectively. There data set was considerably older than our covering years 2013 to 2018. In 2013 the mean national 30-day mortality for hip fracture patients was 8%. In 2019 it was 6.5%. Further we dichotomised the data a sought to determine a difference specifically above and below WHO. Hence the results are not mutually exclusive. Shi et al however intriguingly did observe a possible hermetic effect with ozone, where higher levels of ozone exposure were associated with decreased mortality. Significantly ozone was the only agent in the study where the mean concentration was below the WHO recommended threshold and thus potentially in the hermetic range for mortality.

There is the possibility that the findings are attributable to residual confounder. We consider this unlikely however the sample is large and any systematic confounder would be a mystery. Indeed any confounder would tender to show the opposite from that which we observed. For example increasing pollution is associated with greater population density and urbanisation. It is well established in the UK that urbane living is associated with a high risk of dementia; dementia being a risk factor for post-operative delirium [24]. Again higher pollution levels are associated with poverty and thus inferior health status, which has also been shown to a risk factor for post-operative delirium [25]. Pollution is linked to delirium but only in supra-hormetic exposures [25–27]. Further Across England, Wales and Northern Ireland the care of hip fracture patients is standardised and homogenised via financially incentivised standards of care. Units are the "Best Practice Tariff" only if they comply with all the target criteria. In addition there are comprehensive, explicit and didactic guidelines encompassing almost every aspect of care (https://www.nhfd.co.uk/).

Environmental Targets (Fine Particulate Matter) (England) Regulations 2023 mandate that by the termination of 2040 the mean annual $PM_{2.5}$ level do not exceed 10μg/m$^3$ anywhere in England. This limit is still twice the figure recommended by WHO. The US Environmental Protection Agency recently in January 2024 reset the $PM_{2.5}$ standard from 12 to 9μg/m$^3$ [28] Our findings in this work do not contradict or undermine the rationale that belies the UK and US targets, but neither are they necessarily an endorsement of such policies.

We show that mean annual exposure to the three cardinal air pollutants exceeds WHO safety limit for the majority of hip fracture units with regard to $PM_{2.5}$ and sizeable proportion in relation to $PM_{10}$ and nitrogen dioxide. We observe that exposure does not appear to have an impact on mortality. However there is the possibility of a hermetic effect with regard to post-operative delirium in hip fracture patients. Those treated at units with mean exposures that are low, in absolute terms, but higher than the very lowest exposures experience a lower

incidence of post-operative delirium. Our findings are not unqualified support of a permissive attitude to ambient air pollution. The "price" of the putative benefit on other systems, notably cardiovascular physiological has not been determined. Further hospitals cater for diverse patient populations who may experience responses to pollutants which are divergent to that which has been observed her. It is not often not possible, realistic or desirable to relocate institutions. Nonetheless our findings are pertinent to international, national and regional initiatives seeking to reduce pollution and to consideration as to appropriate, safe and achievable future pollutant targets.

## Author Contributions

**Conceptualization:** Chika Edward Uzoigwe, Rana Muhammad Anss Bin Qadir, Ahmed Daoub.

**Data curation:** Chika Edward Uzoigwe, Rana Muhammad Anss Bin Qadir, Ahmed Daoub.

**Formal analysis:** Chika Edward Uzoigwe, Ahmed Daoub.

**Funding acquisition:** Chika Edward Uzoigwe, Ahmed Daoub.

**Investigation:** Chika Edward Uzoigwe, Rana Muhammad Anss Bin Qadir, Ahmed Daoub.

**Methodology:** Chika Edward Uzoigwe, Rana Muhammad Anss Bin Qadir.

**Project administration:** Chika Edward Uzoigwe, Rana Muhammad Anss Bin Qadir.

**Resources:** Chika Edward Uzoigwe, Rana Muhammad Anss Bin Qadir.

**Supervision:** Chika Edward Uzoigwe, Ahmed Daoub.

**Validation:** Chika Edward Uzoigwe, Ahmed Daoub.

**Visualization:** Chika Edward Uzoigwe, Ahmed Daoub.

**Writing – original draft:** Chika Edward Uzoigwe, Rana Muhammad Anss Bin Qadir, Ahmed Daoub.

**Writing – review & editing:** Chika Edward Uzoigwe, Rana Muhammad Anss Bin Qadir, Ahmed Daoub.

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
