## [Decision Letter · Decision Letter 0]

6 Nov 2024

PONE-D-24-41181Ambient Pollution at Hip Fracture Units and Impact on Mortality and Post-operative Delirium: A Hormetic Effect?PLOS ONE

Dear Dr. Uzoigwe,

Thank you for submitting your manuscript to PLOS ONE. After careful consideration, we feel that it has merit but does not fully meet PLOS ONE’s publication criteria as it currently stands. Therefore, we invite you to submit a revised version of the manuscript that addresses the points raised during the review process.

We look forward to receiving your revised manuscript.

Kind regards,

Pengpeng Ye

Academic Editor

PLOS ONE

Journal requirements: When submitting your revision, we need you to address these additional requirements. 1. Please ensure that your manuscript meets PLOS ONE's style requirements, including those for file naming. The PLOS ONE style templates can be found at https://journals.plos.org/plosone/s/file?id=wjVg/PLOSOne_formatting_sample_main_body.pdf and https://journals.plos.org/plosone/s/file?id=ba62/PLOSOne_formatting_sample_title_authors_affiliations.pdf 2. Please include captions for your Supporting Information files at the end of your manuscript, and update any in-text citations to match accordingly. Please see our Supporting Information guidelines for more information: http://journals.plos.org/plosone/s/supporting-information. 

Reviewers' comments:

Reviewer's Responses to Questions

**Comments to the Author**

1. Is the manuscript technically sound, and do the data support the conclusions?

Reviewer #1: Partly

Reviewer #2: No

2. Has the statistical analysis been performed appropriately and rigorously? 

Reviewer #1: Yes

Reviewer #2: No

3. Have the authors made all data underlying the findings in their manuscript fully available?

Reviewer #1: Yes

Reviewer #2: Yes

4. Is the manuscript presented in an intelligible fashion and written in standard English?

Reviewer #1: Yes

Reviewer #2: Yes

5. Review Comments to the Author

Reviewer #1: The study is very interesting to demonstrate the association between the ambient pollution and health outcomes among hospitals that treating patients with hip fracture in UK and has outputted some primary findings to inform us the importance of reduction of air pollution.

Introduction

1.para No. 1, line 3: a typo “in the long-term”

2.please articulate why choosing hip fracture units to explore the ambient pollution. Authors need to supply the association between hip fracture and air pollution, i.e. why exploring potential causality of air pollution and mortality and delirium.

Methods

1.What is the study design and which reporting guideline authors should follow?

2.Methodology should be improved. The current one is too vague. Did authors consider using multivariable regression to reduce the selection bias through adjusting confounders?

Results

1.A flowchart is needed to demonstrate the enrollment of the participants.

2.Despite a provided declaration regarding limited confounders in discussion, authors might do more to reduce the heterogeneity of hospitals and patients, such as the severity of complications, variation of hospitals’ infrastructure.

3.It seems that authors can do further analysis using UK standards of PM10 or NO2 to identify which thresholds are sensitive to delirium incidence.

Discussions

1.para No.3: two same findings were provided around PM10 and NO2?

2.The example regarding plants cultivation was not appropriate, which should be replaced.

3.page 7: Please rewrite the sentence “Radiation exposures greater than background, but below critical thresholds, do no harm;”

Reviewer #2: Comments:

The study sought to determine the ambient pollution around hip fracture units and its relation to current UK and WHO standards and to outcomes regarding 30-day mortality and post-operative delirium. The topic is very interesting and meaningful. However, the manuscript did not describe a technically sound piece of scientific research with data that supports the conclusions. Statistical analysis was not performed appropriately and rigorously. The Strengthening the Reporting of Observational Studies in Epidemiology (STROBE) reporting guidelines should be followed.

Introduction：

1. The introduction should be more well-founded. The authors should introduce why ambient pollution was considered to have an impact on mortality and post-operative delirium. Existing research results regarding ambient pollution and Impact on these vulnerable and elderly patients and ongoing challenges should be presented.

Methods:

2. The methods part lacks the description of the study population, the clear definition of the exposure and outcomes, the selection of confounding variables, and the clear description of statistical analysis.

3. How was the study population be selected? Selection bias and information bias should be considered.

4. Adjusted model should be added to avoid the confounding bias.

Results:

5. What’s the demographic characteristics of the study population?

Discussion:

6. What’s the strengths and limitations of the study?

6. PLOS authors have the option to publish the peer review history of their article (what does this mean?). If published, this will include your full peer review and any attached files.

Reviewer #1: No

Reviewer #2: No

---

## [Author Response · Author response to Decision Letter 0]

18 Nov 2024

Response to Reviewers

We thank the reviewers for their instructive comments. We have used them to improve the quality of the manuscript

Reviewer #1: The study is very interesting to demonstrate the association between the ambient pollution and health outcomes among hospitals that treating patients with hip fracture in UK and has outputted some primary findings to inform us the importance of reduction of air pollution.

Thank you for your generous comments. We too agree that the submission is of interest.

Introduction

1.para No. 1, line 3: a typo “in the long-term”

Thank you, we apologise for this oversight. A correction has been made.

2.please articulate why choosing hip fracture units to explore the ambient pollution. Authors need to supply the association between hip fracture and air pollution, i.e. why exploring potential causality of air pollution and mortality and delirium.

This is a valid point. We have made it clearer in the manuscript. There is a general belief that pollution is injurious to health, as can be seen by the publications from WHO. We postulated that it may be so even in an acute setting. However, if this is indeed so, it will be most demonstrable in the most vulnerable, hence in those with the most compromised physiological reserve. The youth will have little short-term out deleterious outcomes with regard to mortality or cognition in the face of the ambient pollution due to their capacious physiological reserve. The situation is materially different for those of advanced aged, especially if they have already suffered an injury. Hence, while the young and healthy are unlikely to suffer from labile pollution levels, the same cannot be said for the elderly. This is fundamentally reason the pollution guidelines are so important. They are there to protect the most vulnerable from harm. If we focus on the general population, it may seem as though such regulations are excessive. In order to see if the WHO guidance is warranted, we must explore the impact of pollution on those most susceptible to their harms. 

Methods

1.What is the study design and which reporting guideline authors should follow?

Thank you for this question. It is a cross-sectional observation study. The Strengthening the Reporting of Observational Studies in Epidemiology model was followed. We have made both points more explicitly clear.

2.Methodology should be improved. The current one is too vague. Did authors consider using multivariable regression to reduce the selection bias through adjusting confounders?

This is an excellent question. There is no individual data but rather group data. The sample data is exceedingly large so we did not anticipate any systemic error. Secondly, and possibly most importantly, the care of hip fracture patients in the UK is standardised and homogenised. There are explicit and didactic guidelines relating to virtually every aspect of their care (https://www.nhfd.co.uk/). Further institutional compliance is not optionally but rather financially incentivised. Indeed hospitals must satisfy all the care standards if they are to be awarded what is termed the Best Practice Tariff. In addition there are centralised remedial measures for units that are outliers, with an explicit “Outlier Policy”. In these circumstances we decided against statistical regression adjustment. We have clarified and made more explicit our methods in an evidence base manner.

Results

1.A flowchart is needed to demonstrate the enrollment of the participants.

This is a valid point and has been added, showing institutional enrolment, as we used institutional level data.

2.Despite a provided declaration regarding limited confounders in discussion, authors might do more to reduce the heterogeneity of hospitals and patients, such as the severity of complications, variation of hospitals’ infrastructure.

This is an excellent point the reviewer raises. We have included the discussion above relating the implementation of the Best Practice Tariff and homogenisation of care. The sample sizes are very large leading us not to suspect such pervasive systematic confounders.

3.It seems that authors can do further analysis using UK standards of PM10 or NO2 to identify which thresholds are sensitive to delirium incidence.

This is a valid and insightful observation; however it falls out the remit of our initial study plan. It is definitely the foundation of a future study.

Discussions

1.para No.3: two same findings were provided around PM10 and NO2?

We are grateful to the reviewer for observing this oversight on our part. It is a transcription error. The figures reported in the methods and in the tables are correct. The text has now been transcribed correctly.

2.The example regarding plants cultivation was not appropriate, which should be replaced.

We appreciate how this may seem misplaced. However we feel very strongly that it is important to show that the concept of hormesis did not appear in abstraction. We therefore discussed the origin of the phenomenon. It may be new to many readers. Our aim is to show that it is an atavistic deeply-engrained evolutionary phenomenon, found in the very earliest and simplest life-forms. It is not a feature that has appeared in the 21st century that is exclusive to humans. As it is of such an empiric and fundamental origin it will be observed in humans and have potential healthcare ramifications and manifestations. 

3.page 7: Please rewrite the sentence “Radiation exposures greater than background, but below critical thresholds, do no harm;”

This is valid and has been re-composed for clarity.

Reviewer #2: Comments:

The study sought to determine the ambient pollution around hip fracture units and its relation to current UK and WHO standards and to outcomes regarding 30-day mortality and post-operative delirium. The topic is very interesting and meaningful. 

Thank you. This is a sound synopsis of our work. We appreciate your comments that the work is interesting and meaningful.

However, the manuscript did not describe a technically sound piece of scientific research with data that supports the conclusions. 

Our opinions differ from those of the review on this point. We look at the annual mortality and prevalence of delirium for hip fracture patients in every hospital in England, Wales and Northern Ireland. We see if there is a correlation with annual ambient pollution and these outcomes. Our samples sizes are massive. In the UK the care of hip fracture patients is homogenised via the mandatory of use financially incentivised explicit, didactic and comprehensive guidelines (https://www.nhfd.co.uk/). This helps to eradicate biases. There are centralised policies for the units that are outliers, with an express “Outlier Policy”.

We observe no impact on mortality or delirium. However there is the additional intriguing phenomenon where by low-level sub-injurious levels of certain pollutants may be protective against delirium. Our findings tessellate and correlate with those of numerous of the researchers who have observed this effect with regard to pollution but been unable to account for it. The role of the study in evidence tapestry is critical. WHO have requested certain target pollution levels. Individual nations have opted for different and higher limits. Our study shows that even in the most vulnerable settings, such as hospitals with elderly hip fracture patients, the WHO target levels are exceeded. However there does not appear to be an increase in mortality or delirium in UK associated with treatment in units located at pollutant levels exceeding WHO thresholds. 

We have considered the reviewer’s insight in this area and adapted the article to make these finding and how they reached more conspicuous.

Statistical analysis was not performed appropriately and rigorously. 

Our opinion must again diverge from the reviewer. Chi Squared is the appropriate test. The ample sample size and standardised care operated to help mitigate bias. We accept no single method is infallible. We have considered the comments and clarified our position in the manuscript. 

We are also mindful that the reviewer does raise an intriguing and important point. Given the fact that multiple statistical tests are performed there is the possibility some maybe stochastically positive. We did consider performing a Bonferoni correction to account for this. However our p values are exceeding low, below 0.0001, such that we did not feel it meaningful to engage in this line of discussion. Had our p values been borderline, in the region of 0.01, this may have been of some merit.

The Strengthening the Reporting of Observational Studies in Epidemiology (STROBE) reporting guidelines should be followed.

Thank you for this comment. This reporting paradigm has been followed. We have made it more evidently so in the manuscript.

Introduction：

The introduction should be more well-founded. The authors should introduce why ambient pollution was considered to have an impact on mortality and post-operative delirium. Existing research results regarding ambient pollution and Impact on these vulnerable and elderly patients and ongoing challenges should be presented.

We thank the reviewer for this excellent observation. There is a paucity of the information referred to by the reviewer in our introduction simply because our study is one of the first, if not the first to explore these relationships, especially in a senescent hospitalised cohort. However the topic is valid and meritorious of investigation. It is intuitive that pollution may be injurious to cognition especially in those of a more advanced age and thus with potentially a more constrained cognitive reserve. Nonetheless we have explored and added more references.

Methods:

2. The methods part lacks the description of the study population, the clear definition of the exposure and outcomes, the selection of confounding variables, and the clear description of statistical analysis.

We regret that this is not clear. It has been clarified.

3. How was the study population be selected? Selection bias and information bias should be considered.

We are grateful again to the review for this insightful question. This is indeed one of the greatest strengths of our study. There was no selection; every hospital in the England, Wales and Northern Ireland treating elderly hip fracture patients was included. 

4. Adjusted model should be added to avoid the confounding bias.

Thank you for this recommendation. However there is group data rather than individual data. Hence we look at annual institutional mortality and delirium of hospitals. We performed neither selection nor de-selection of hospitals. The care of hip fracture patients in the England, Wales is standardised and homogenised. There are mandatory guidelines which are financially incentivised. Units only receive this financial remuneration if they comply with all the measures (The National Hip Fracture Database (nhfd.co.uk)). We are minded that the sample size of the grouped data and homogenisation of care will mitigate against bias.

Results:

5. What’s the demographic characteristics of the study population?

Thank you, we regret that this is not clear. They are hip fracture age 65 years are older. There are no other selection criteria. This is cohort, due their senescence and reduced physiological reserve are most vulnerable to ambient pollution. This is one of the groups any such guidelines are designed to protect, namely the most susceptible. 

Discussion:

6. What’s the strengths and limitations of the study?

The reviewers have astutely identified some of the weaknesses in the study which have incorporated and addressed.

We thank the reviewers again for the incisive observations and recommendations.

---

## [Decision Letter · Decision Letter 1]

2 Dec 2024

Ambient Pollution at Hip Fracture Units and Impact on Mortality and Post-operative Delirium: A Hormetic Effect?

PONE-D-24-41181R1

Dear Dr. Uzoigwe,

We’re pleased to inform you that your manuscript has been judged scientifically suitable for publication and will be formally accepted for publication once it meets all outstanding technical requirements.

Kind regards,

Pengpeng Ye

Academic Editor

PLOS ONE

Additional Editor Comments (optional):

Reviewers' comments:

Reviewer's Responses to Questions

**Comments to the Author**

1. If the authors have adequately addressed your comments raised in a previous round of review and you feel that this manuscript is now acceptable for publication, you may indicate that here to bypass the “Comments to the Author” section, enter your conflict of interest statement in the “Confidential to Editor” section, and submit your "Accept" recommendation.

Reviewer #1: All comments have been addressed

2. Is the manuscript technically sound, and do the data support the conclusions?

Reviewer #1: Yes

3. Has the statistical analysis been performed appropriately and rigorously? 

Reviewer #1: Yes

4. Have the authors made all data underlying the findings in their manuscript fully available?

Reviewer #1: Yes

5. Is the manuscript presented in an intelligible fashion and written in standard English?

Reviewer #1: Yes

6. Review Comments to the Author

Reviewer #1: Thanks for your justification. I satisfied the comprehensive and insightful answers authors provided. I have no further comments.

7. PLOS authors have the option to publish the peer review history of their article (what does this mean?). If published, this will include your full peer review and any attached files.

Reviewer #1: **Yes: **Jing Zhang

---

## [Editor Report · Acceptance letter]

6 Dec 2024

PONE-D-24-41181R1 

PLOS ONE

Dear Dr. Uzoigwe, 

I'm pleased to inform you that your manuscript has been deemed suitable for publication in PLOS ONE. Congratulations! Your manuscript is now being handed over to our production team.

Kind regards, 

on behalf of

Dr. Pengpeng Ye 

Academic Editor

PLOS ONE